# Abdominal Multi-organ Segmentation using CNN and Transformer

Rui Xin[1][0000−0002−0224−3431] and Lisheng Wang[1][0000−0003−3234−7511]

Institute of Image Processing and Pattern Recognition, Department of Automation,
Shanghai Jiao Tong University, China
`lswang@sjtu.edu.cn`

**Abstract.** In this paper, we combine the advantages of convolution local correlation and translation invariance in CNN with Transformer's ability to effectively capture long-term dependencies between pixels to produce high-quality pseudo labels. In order to segment images efficiently and quickly, we select nnU-Net [2] as the final segmentation network and use pseudo labels, unlabeled data and labeled data together to train the network, and then we use Generic U-Net [2], the backbone network of nnU-Net, as final prediction network. The mean DSC of the prediction results of our method on validation set of FLARE2022 Challenge [3] is 0.7580.

**Keywords:** Medical segmentation · Pseudo label · Semi-supervision learning.

## 1 Introduction

Accurate segmentation of organs or lesions from medical images plays an important role in many clinical applications, such as diagnosis, treatment and post-operative planning. With the increase of annotation data, deep learning has achieved great success in image segmentation. However, for medical images, the acquisition of annotation data is often expensive because of the expertise and time required to generate accurate annotations, especially in 3D images.

In order to reduce labeling cost, many methods have been proposed in recent years to develop high-performance medical image segmentation models to reduce labeling data. A small amount of labelled data and a large amount of unlabeled data are more consistent with the actual clinical scenarios. The semi-supervised learning framework obtains high-quality segmentation results by learning directly from limited labeled data and a large amount of unlabeled data.

In this paper, a semi-supervised method for abdominal multi-organ image segmentation is proposed, which combines CNN and Transformer [1] to generate a large amount of pseudo labels, and uses pseudo labels, unlabeled data and labeled data to train the network, which is equivalent to dataset augmentation and improving the performance of the network.

## 2    Method

This chapter focuses on two network frameworks used to generate high-quality pseudo labels, and the entire process of using the pseudo label to improve the performance of the backbone network.

### 2.1    nnU-Net

**Preprocessing**  We first crop the non-zero regions of the image and resample the cropped data, and then we use Z-Score standardization to normalize the data. The Z-Score standardized formula is as follows:

$$z = \frac{x - \mu}{\sigma} \tag{1}$$

$\mu$ is the average value of the CT value of the image label, $\sigma$ is the variance of the CT value of the image label.

**Network**  We use 3D U-Net [8] at full resolution for training. As shown in Figure 1, this 3D U-Net is Generic U-Net, the backbone network of nnU-Net, which is also used as the final prediction network.

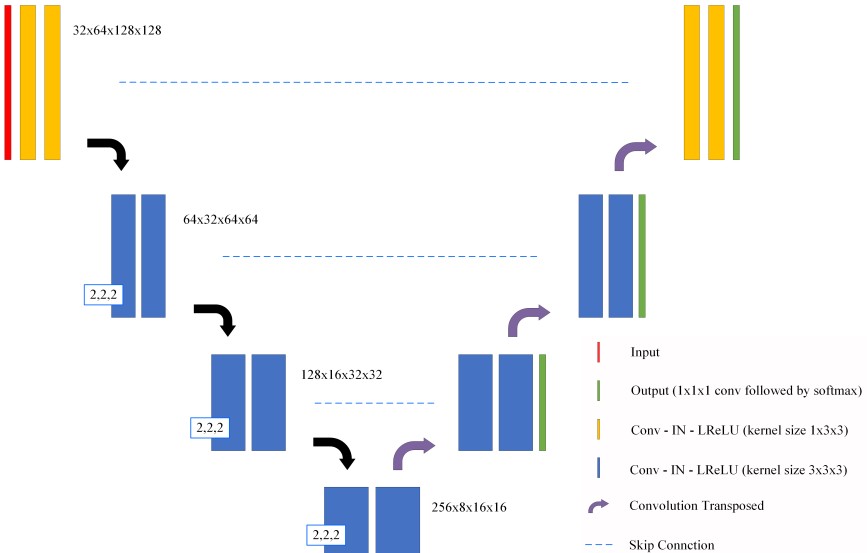

**Fig. 1.** Generic U-Net, the backbone network of nnU-Net.

**Training**  We use the sum of dice loss and cross entropy loss as our total loss function:

$$\mathcal{L}_{\text{total}} = \mathcal{L}_{\text{dc}} + \mathcal{L}_{\text{ce}} \tag{2}$$

The dice loss function is formulated as follows:

$$\mathcal{L}_{\text{dc}} = -\frac{2}{|K|} \sum_{k \in K} \frac{\sum_{i \in I} u_i^k v_i^k}{\sum_{i \in I} u_i^k + \sum_{i \in I} v_i^k} \tag{3}$$

Where u is softmax output and v is one hot encoding ground and truth. K is the number of categories. The formula of cross entropy loss function is as follows:

$$\mathcal{L}_{\text{ce}} = -\sum_{x} p(x) \log q(x) \tag{4}$$

The probability distribution p is the expected output, and the probability distribution q is the actual output.

**Testing**  The whole testing process is based on the patch size and we use TTA for data augmentation.

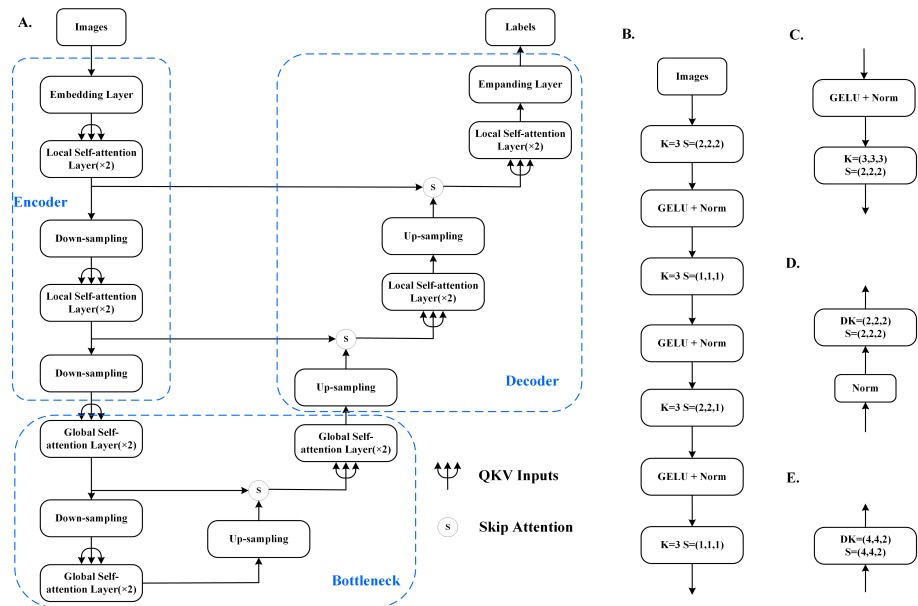

**Fig. 2.** The overall structure of nnFormer. A shows the architecture of nnFormer. B, C, D and E are the specific details of embedding layer, down-sampling layer, up-sampling layer and expanding layer, respectively. K represents the convolutional kernel size. S represents the stride. DK represents the deconvolutional kernel size. Norm is the normalization strategy.

## 2.2   nnFormer [9]

**Network**  As shown in Fig. 2, the backbone structure of nnFormer is mainly composed of encoder, bottleneck layer and decoder.

The encoder includes an embedding layer, two local self-attention layer blocks and two down-sampling layers. The input image is transformed into features that can be processed by the network through the convolution structure.

The decoding part symmetrically includes two local self-attention layer blocks, two up-sampling layers and the last patch expanding layer for mask prediction. nnFormer uses a local 3D image block-based self-attention calculation called V-MSA [9]. Compared with the traditional voxel self-attention calculation method, V-MSA can greatly reduce the computational complexity.

The bottleneck layer consists of a down-sampling layer, an up-sampling layer, and three global self-attention layer blocks to provide a large receive domain to support the decoder. At the same time, adding skip attention [9] connections in a symmetrical manner between the corresponding feature pyramids of the encoder and decoder helps to recover fine-grained details in the prediction.

**Training and testing**  In nnFormer, we use the same training and testing strategy as nnU-Net.

## 2.3   Proposed method

The overall architecture of the approach is shown in Fig. 3, which consists of the generation of pseudo label and the prediction network. In pseudo label generation stage, nnU-Net and nnFormer network models are mainly used. In the final prediction part, we adopte Generic U-Net, the basic network model of nnU-Net.

**Pseudo label generation**  Specifically, in the generation stage of pseudo label, we mainly adopt two network models, nnU-Net and nnFormer. We first train the two models with only 50 cases of labeled data, and then predicted the unlabeled data respectively, and generated the final prediction result by means of prediction probability fusion. This method combine the advantages of local correlation of convolution to spatial information encoding in CNN and long-term dependency capturing in Transformer [4].

After the prediction results are obtained, we use the connected domain analysis for data selection, only the largest part of the connected domain results of each label were saved. Finally, the pseudo label containing each organ is obtained, as shown in Fig. 4. We use ITK-SNAP [7] for visualization.

**Predictive network**  To improve the segmentation efficiency, we use simple network structure for final prediction. We adopt the backbone network Generic U-Net in nnU-Net method as our predictive network. After obtaining pseudo label, the original label and generated pseudo label are trained through nnU-Net, and finally Generic U-Net, the basic network of nnU-Net, is used as the final prediction network.

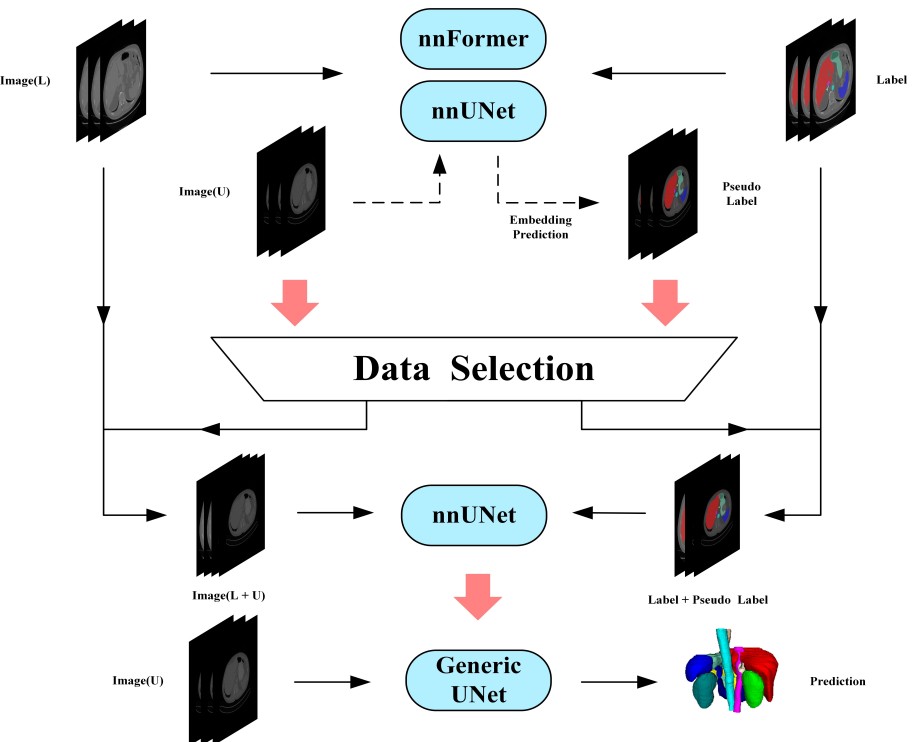

**Fig. 3.** The overall architecture. Images(L) represents the labeled image. Images(U) represents the unlabeled image. Images(L+U) represents the labeled and unlabeled image mixed together.

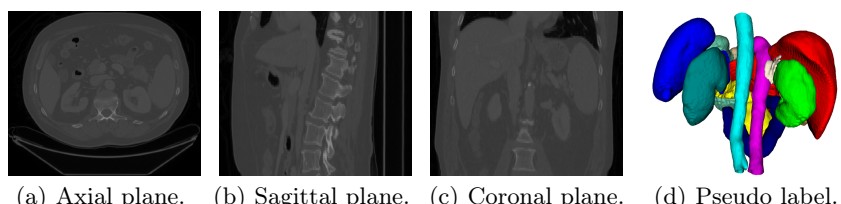

(a) Axial plane.     (b) Sagittal plane.    (c) Coronal plane.     (d) Pseudo label.

**Fig. 4.** Three planes of an unlabeled CT image and corresponding generated pseudo label.

**Post-processing** In some computer vision tasks, it is necessary to do some post-processing on the output of the model to optimize the visual effect, and connected domain is a common post-processing method. Especially for segmentation tasks, sometimes there are some false positives in the output mask. Finding independent contours with small area through 3D connected domain and removing them can effectively improve the visual effect. We use connected domain principal component analysis to remove 3D small connected domains and retain the largest part of each label connected domain.

## 3   Experiments

**Dataset** The FLARE 2022 is an extension of the FLARE 2021 [5] with more segmentation targets and more diverse abdomen CT scans. The FLARE2022 Challenge [3] provides a small amount of labeled cases and a large amount of unlabeled cases regarding abdominal organs. The training set includes 50 labeled CT images and organ of patients with pancreatic disease and 2000 unlabeled CT images of patients with pancreatic disease. The organs to be segmented include 13 organs, including liver, spleen, pancreas, right kidney, left kidney, stomach, gallbladder, esophagus, aorta, inferior vena cava, right adrenal gland, left adrenal gland and duodenum. The validation set includes 50 CT images from patients with liver, kidney, spleen, or pancreas disease. The test set includes 100 CT images of patients with liver, kidney, spleen, and pancreas diseases and 100 CT images of patients with endometrial, bladder, stomach, sarcoma, and ovarian diseases. [6]

**Evaluation measures** The evaluation indexes of this competition include dice similarity coefficient, normalized surface dice, running time, area under GPU memory time curve and area under CPU utilization time curve.

The dice similarity coefficient is a statistic used to evaluate the similarity of two samples, essentially measuring the overlap of two samples. The formula is as follows:

$$\mathcal{D}SC = \frac{2|X \cap Y|}{|X| + |Y|} \tag{5}$$

$|X|$ and $|Y|$ represent the number of elements in each set, respectively. It is used to measure how similar the prediction result is to the original label. Normalized surface dice is a boundary-based evaluation method used to describe the boundary error between the prediction result and the original label. In addition, the GPU memory and GPU utilization are recorded every 0.1s, and the area under the GPU memory-time curve and the area under the CPU utilization-time curve are cumulative values of running time.

**Implementation details** The development environments and requirements are presented in Table 1.

**Table 1.** Development environments and requirements.

| | |
|---|---|
| Ubuntu version | Ubuntu 18.04.5 LTS |
| CPU | Intel(R) Core(TM) i9-10920X CPU@3.50GHz |
| RAM | 126 GB |
| GPU | 1 NVIDIA GeForce RTX 3090(24G) |
| CUDA version | 11.4 |
| Programming language | Python 3.6 |
| Deep learning framework | Pytorch (Torch 1.9.1, torchvision 0.10.1) |

**Table 2.** Training protocol.

| | |
|---|---|
| Batch size | 2 |
| Patch size | $64 \times 128 \times 128$ |
| Total epochs | 1000 |
| Optimizer | SGD with nesterov momentum($\mu = 0.99$) |
| Initial learning rate(lr) | 0.01 |
| Network initialization | "he" normal initialization |
| Lr decay schedule | "poly" strategy 6 |
| Training time | 90 hours |
| Loss function | Sum of cross entropy loss and dice loss |

We use the same training strategy for nnU-Net and nnFormer. The training protocol is presented in Table 2.

Before training, we resample all images to the same spacing. In the process of training, we use data augmentation methods such as rotation, scaling, Gaussian noise, Gaussian blur, gamma enhancement and mirror image.

$$lr = initial\_lr \times \left(1 - \frac{epoch\_id}{max\_epoch}\right)^{0.9} \qquad (6)$$

## 4   Results

### 4.1   Quantitative results on validation set

This method combines the advantages of CNN and Transformer to produce a high-quality pseudo label. We use 50 labeled data for training and test on the validation set. This produces a higher quality result than using either model alone, and their respective dice score metrics on the validation set are shown in Table 3.

We compare the prediction results of this method with those of directly transferring to Generic U-Net after training without using pseudo label. The dice score of the predicted results on the validation set without and with pseudo label training are shown in Table 4. The results show that using pseudo label can greatly improve network segmentation performance.

**Table 3.** The dice metrics of the prediction results of nnU-Net and nnFormer on the validation set and The dice metrics of their embedding prediction results on the validation set. RK, IVC, RAG, LAG and LK represent right kidney, inferior vena cava, right adrenal gland, left adrenal gland and left kidney respectively.

| Methods | Average | Liver | RK | Spleen | Pancreas | Aorta | IVC | RAG |
|---|---|---|---|---|---|---|---|---|
| nnU-Net | 0.8310 | 0.9512 | 0.8640 | 0.8734 | 0.8494 | 0.9377 | 0.8747 | 0.7972 |
| nnFormer | 0.8008 | 0.9550 | 0.7936 | 0.8867 | 0.8244 | 0.9208 | 0.8093 | 0.7358 |
| nnU-Net+nnFormer | 0.8500 | 0.9631 | 0.8926 | 0.9080 | 0.8709 | 0.9475 | 0.8674 | 0.8049 |

| Methods | LAG | Gallbladder | Esophagus | Stomach | Duodenum | LK |
|---|---|---|---|---|---|---|
| nnU-Net | 0.7820 | 0.6456 | 0.8079 | 0.8502 | 0.7164 | 0.8530 |
| nnFormer | 0.7383 | 0.8441 | 0.7308 | 0.8562 | 0.6467 | 0.8189 |
| nnU-Net+nnFormer | 0.7807 | 0.7168 | 0.8185 | 0.8992 | 0.7363 | 0.8441 |

**Table 4.** The dice metrics of the prediction results of this method and using Generic U-Net directly without pseudo label on the validation set. RK, IVC, RAG, LAG and LK represent right kidney, inferior vena cava, right adrenal gland, left adrenal gland and left kidney respectively.

| Methods | Average | Liver | RK | Spleen | Pancreas | Aorta | IVC | RAG |
|---|---|---|---|---|---|---|---|---|
| w/ pseudo label | 0.7580 | 0.9540 | 0.7972 | 0.8265 | 0.6980 | 0.9233 | 0.8662 | 0.6560 |
| w/o pseudo label | 0.6376 | 0.8891 | 0.6112 | 0.7384 | 0.6207 | 0.8285 | 0.7015 | 0.5012 |

| Methods | LAG | Gallbladder | Esophagus | Stomach | Duodenum | LK |
|---|---|---|---|---|---|---|
| w/ pseudo label | 0.6276 | 0.6547 | 0.7334 | 0.7333 | 0.6012 | 0.7832 |
| w/o pseudo label | 0.5089 | 0.5529 | 0.6184 | 0.6829 | 0.4356 | 0.6004 |

This shows the value of large amounts of unlabeled data. A large amount of unlabeled image data is used to generate pseudo labels, which can get high-quality data after selection, which can make up for the shortage of labels to some extent and improve the prediction ability of the model. For prediction on the validation set, some results and their corresponding labels are shown in Fig. 5. The structure of prediction network is simple and it is difficult to learn deeper features, so the prediction results of some unseen CT images are bad.

### 4.2   Segmentation efficiency results on validation set

In this paper, we adopt Generic U-Net, the backbone network of nnU-Net, as the final prediction network. Because the size of some images is too large, nnU-Net or nnFomer consumes too much RAM, which exceeds the required maximum limit. nnU-Net or nnFomer can not be used as the final predictive framework. Compared with nnU-Net or nnFormer, this method can greatly reduce RAM,

**Table 5.** The average efficiency results of this method on validation set. The efficiency results include running time, maximum memory consumed by GPU, area under GPU memory-time curve and area under CPU utilization-time curve.

| Time | GPU(Max Memory) | AUC(GPU-Time) | AUC(CPU-Time) |
|------|-----------------|---------------|---------------|
| 111.18 $s$ | 2433 $MiB$ | 256444.39 $MiB \times s$ | 2023.45 $MiB \times s$ |

**Table 6.** The DSC index of the results on final testing set. RK, IVC, RAG, LAG and LK represent right kidney, inferior vena cava, right adrenal gland, left adrenal gland and left kidney respectively.

|  | Liver | RK | Spleen | Pancreas | Aorta | IVC | RAG | LAG |
|------|-------|------|--------|----------|-------|------|------|------|
| AVG | 0.9360 | 0.7050 | 0.7471 | 0.6114 | 0.8698 | 0.8314 | 0.6374 | 0.5206 |
| STD | 0.0636 | 0.3774 | 0.3356 | 0.2760 | 0.2081 | 0.1788 | 0.2840 | 0.3357 |

|  | Gallbladder | Esophagus | Stomach | Duodenum | LK |
|------|-------------|-----------|---------|----------|------|
| AVG | 0.5574 | 0.6480 | 0.5956 | 0.4902 | 0.6727 |
| STD | 0.4052 | 0.2592 | 0.3648 | 0.2943 | 0.3953 |

GPU memory consumption and running time because of the simple predictive framework. The efficiency indicators in the validation set are shown in Table 5. Beyond that, we do not optimize the segmentation efficiency.

### 4.3   Results on final testing set

The DSC index of the results on final testing set is shown in Table 6, and the NSD index of the results on final testing set is shown in Table 7.

### 4.4   Limitation and future work

The method proposed in this paper only adopts CNN in the final prediction network, and the limited receptive field leads to the failure to capture global information. In the future, it is hoped to design a lightweight network combining the characteristics of CNN and Transformer for efficient inference of images.

## 5   Conclusion

In this paper, we combine the advantages of CNN and Transformer to establish a long-term dependency relationship, and produce high-quality pseudo labels to enhance the performance of network segmentation. Moreover, we adopt Generic U-Net, the backbone network of nnU-Net, as the final prediction network. The results show that the combination of the two methods produce a high-quality pseudo label compared to using CNN or Transformer alone, and the method

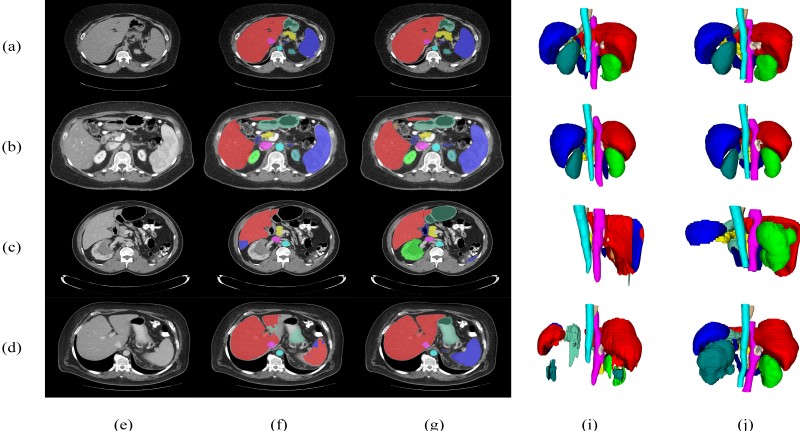

**Fig. 5.** Some visualized results on the validation set and corresponding labels. Row (a) and row (b) are good predicted results and corresponding labels. Row (c) and row (d) are bad predicted results and corresponding labels. Column (f) are predicted results of axial slices and column (g) are corresponding labels of axial slices. Column (i) are 3D results of predicted results and column (j) are 3D results of corresponding labels.

**Table 7.** The NSD index of the results on final testing set. RK, IVC, RAG, LAG and LK represent right kidney, inferior vena cava, right adrenal gland, left adrenal gland and left kidney respectively.

|     | Liver | RK | Spleen | Pancreas | Aorta | IVC | RAG | LAG |
|-----|-------|------|--------|----------|-------|-------|--------|--------|
| AVG | 0.8907 | 0.6958 | 0.7239 | 0.6887 | 0.8779 | 0.8259 | 0.7366 | 0.6043 |
| STD | 0.1183 | 0.3688 | 0.3447 | 0.2844 | 0.2150 | 0.1841 | 0.2995 | 0.3684 |

|     | Gallbladder | Esophagus | Stomach | Duodenum | LK |
|-----|-------------|-----------|---------|----------|--------|
| AVG | 0.5290 | 0.7430 | 0.6039 | 0.6410 | 0.6752 |
| STD | 0.4085 | 0.2683 | 0.3581 | 0.3088 | 0.3849 |

achieves effective semi-supervised segmentation performance in the FLARE2022 Challenge.

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
