# OpenReview forum: "Abdominal Multi-organ Segmentation using CNN and Transformer"
_MICCAI.org/2022/Challenge/FLARE_

### Official Review · Reviewer_Jdin · 2022-09-13
**This paper has sufficient sections and provides detailed information for each model, loss functions and configuration they used. On top of that, they report adequate quantitative and qualitative results although some comments should be given for these results.**

**Rating:** 6
**Confidence:** 4

**Review:**

The paper adopts both nnUnet and nnFormer for training and pseudo-labels generation tasks. Then another simpler Generic Unet is trained on the original labeled data and pseudo ones.

This paper has sufficient sections and provides detailed information for each model, loss functions and configuration they used. On top of that, they report adequate quantitative and qualitative results although some comments should be given for these results.

Having said that, reviewer does not see any significant or prominent contribution of the authors in this work since they mostly just reuse previous works.

Minor suggested improvements:
- Figure 3 should point out both good and bad cases, and should be visualized in settings (recommended by the organizer)
- Subjective comments should be given on the reported evaluation results

---

> ### Author Response · Authors · 2022-10-20
> **Response from Authors**
>
> Thanks for your review and suggestion. In the future, we plan to adopt other lightweight methods or develop some lightweight modules to improve network performance. In the latest revision, the visualization and description of the good cases and bad cases, as well as the subjective comments on the evaluation results are included.

---

### Official Review · Reviewer_FPEj · 2022-09-16
**good work but talk about the networks little.**

**Rating:** 7
**Confidence:** 3

**Review:**

This work use nnUNet and nnFormer to generate the pseudo labels, but i can't find the structure or description of the networks, maybe they use the default model.
Since the nnUnet or nnFormer consume  too much RAM, which exceeds the required maximum
limit.It makes me confused that the author adopt the generic UNet in nnUNet as the predictive model to reduce the RAM, GPU consume and the running time.
It will be better to talk about details of the adopted generic Unet.

---

> ### Author Response · Authors · 2022-10-20
> **Response of Authors**
>
> Thanks for your review and suggestion. In the latest revision, we have added a description of the model structure, and we have added an detailed introduction to the predictive network model Generic U-Net.

---

### Official Review · Reviewer_DRuo · 2022-09-19
**May consider further improvement for the efficiency of both training and inference**

**Rating:** 8
**Confidence:** 3

**Review:**

Pros: 1. Unlabeled data is well used by weak-supervised training which has a ~8% boost in DSC. 2. Combining nnUNet and nnFormer has gained a ~2% boost in DSC compared to nnUNet only.

Cons: 1. May consider further improvement for the efficiency of both training and inference.

---

> ### Author Response · Authors · 2022-10-20
> **Response of Authors**
>
> Thanks for your review and suggestion. In the future, we plan to make some lightweight modules to improve the accuracy of segmentation, while not affecting the computing resources occupied by the whole model.

---

### Meta-Review · Program_Chairs · 2022-09-28

**Recommendation:** Major Revision
**Confidence:** 5

**Metareview:**

Reviewers raise many concerns and suggestions. Please address all comments in the revised manuscript.